# Searching for the Best Values of NMR Shielding and Spin-Spin Coupling Parameters: CH_4-n_F_n_ Series of Molecules as the Example

**DOI:** 10.3390/molecules28031499

**Published:** 2023-02-03

**Authors:** Karol Jackowski, Mateusz A. Słowiński

**Affiliations:** Laboratory of NMR Spectroscopy, Faculty of Chemistry, University of Warsaw, Pasteura 1, 02-093 Warsaw, Poland

**Keywords:** NMR spectroscopy, fluoromethanes, nuclear magnetic shielding, spin-spin coupling

## Abstract

Attempts at the theoretical interpretation of NMR spectra have a very long and fascinating history. Present quantum chemical calculations of shielding and indirect spin-spin couplings permit modeling NMR spectra when small, isolated molecules are studied. Similar data are also available from NMR experiments if investigations are performed in the gas phase. An interesting set of molecules is formed when a methane molecule is sequentially substituted by fluorine atoms—CH_4-n_F_n_, where *n* = 0, 1, 2, 3, or 4. The small molecules contain up to three magnetic nuclei, each with a one-half spin number. The spectral parameters of CH_4-n_F_n_ can be easily observed in the gas phase and calculated with high accuracy using the most advanced ab initio methods of quantum chemistry. However, the presence of fluorine atoms makes the calculations of shielding and spin-spin coupling constants extremely demanding. Appropriate experimental ^19^F NMR parameters are good but also require some further improvements. Therefore, there is a real need for the comparison of existing NMR measurements with available state-of-the-art theoretical results for a better understanding of actual limits in the determination of the best shielding and spin-spin coupling values, and CH_4-n_F_n_ molecules are used here as the exceptionally important case.

## 1. Introduction

Nuclear magnetic moments have been known since Rabi’s molecular beam experiments [1,2], which were performed before World War II. Later, in 1945, two research groups in the USA independently discovered nuclear magnetic resonance (NMR) in macroscopic samples [3,4]. A new physical method was considered a useful tool for the determination of nuclear magnetic moments though its results were dependent on selected samples [5]. In 1950, Proctor and Yu [6] proved the existence of ^14^N chemical shifts for NH_4_^+^ and NO_3_^−^ ions. Independently, Dickinson [7] found ^19^F chemical shifts for a range of fluorine-containing compounds, and both the above publications appeared in the same issue of *Physical Review*. The chemical shifts are due to the magnetic shielding of nuclei by electrons, and this phenomenon was theoretically described first by Ramsey [8,9,10]. Meanwhile, the indirect interactions of nuclear magnetic moments were also found and carefully analyzed [11,12,13]. A new analytical method of chemical compounds was born and fast became recognized as very important for chemistry [14] although at the beginning, it was still limited to the observation of nuclei with high natural abundances such as protons or fluorine-19.

Small organic molecules containing halogen atoms (F, Cl, Br, or I) have been extensively used in nuclear magnetic resonance (NMR) spectroscopy for over 70 years. Nowadays, such liquids are mostly applied as solvents for NMR experiments, e.g., tetrachloromethane (CCl_4_), chloroform (CHCl_3_), or deuterated chloroform (CDCl_3_), but 1,1,1-trichloro-2,2,2-trifluoroethane (C_2_F_3_Cl_3_, also known as freon 113) was among other fluorine compounds selected for the first observation of ^19^F NMR chemical shifts [7] and corresponding medium effects [15]. From the beginning, ^1^H and ^19^F NMR spectra were immediately adopted for the identification of organic compounds, but the first measurements of proton and fluorine chemical shifts were often unstable and dependent on the selected solvent. Consequently, halogen-containing solvents were also applied for the studies of medium effects in chemical shifts [16,17,18,19,20]; examples of such early investigations are given in the Pople, Schaefer, and Schneider monography [14], and an early review of liquid and gaseous results was completed by Rummens [21]. ^13^C spectra were available a bit later because of the low natural abundance of carbon-13 isotope [22], and the appropriate medium effects in ^13^C NMR were successfully studied when the Fourier-transform (FT) technique [23] was applied to standard NMR spectrometers. The early ^13^C measurements of shielding and spin-spin coupling in gaseous fluoromethanes were performed in 1977 [24].

Let us note that halogen-containing molecules were also used for the standardization of proton chemical shifts, e.g., CHCl_3_ in Buckingham et al.’s work [20]. This idea was slightly changed when Thiers suggested using tetramethylsilane (TMS, (CH_3_)_4_Si) for internal referencing of proton chemical shifts [25]. TMS as the reference standard remains obligatory also today with the modification that deuterated chloroform (CD_3_Cl) is the solvent for the measurements of ^1^H, ^13^C, and ^29^Si NMR chemical shifts [26,27]. Generally, all the chemical shifts (δ_i_) are determined relative to the shielding of a selected reference standard, and the chemical shift is expressed as follows:(1)$$\delta_{i}=\;\left(\sigma_{\mathrm{ref}}{\;-\;\sigma }_{i}\right)/\left({1\;-\;\sigma }_{\mathrm{ref}}\right),$$
where σ_ref_ and σ_i_ are the shielding constants of the observed nuclei in reference and investigated compounds, respectively. δ_i_ is usually multiplied by 10^6^ and given in parts per million (ppm). Earlier, a slightly different scheme was applied for the referencing of ^1^H chemical shifts, known as the *τ-scale* [28]. In the *τ-scale,* the TMS signal was assigned exactly 10.00 ppm, ^1^H chemical shifts were defined as τ_i_ = 10.00—σ_i_, and the ^1^H measurements were performed for diluted solutions in CCl_4_. Actually, the *τ-scale* convention is not in use, but this concept should be remembered for all older measurements of chemical shifts that were defined in the opposite direction than δ_i_ values: in the actual scale, δ_i_ values increase with the decrease of magnetic shielding, and previously, chemical shifts and shieldings have been defined in the same direction [14].

Equation (1) contains the shielding parameter (σ), which is dependent on the electronic structure of molecules and can be calculated using approximate methods of quantum chemistry [29,30,31]. It is the second-rank tensor, but due to the free reorientation of molecules in gases and liquids, we observe only the averaged σ value of shielding, often called the shielding constant. In fact, the σ parameter is never constant, as it is dependent on many factors such as variable medium effects or temperature. However, this colloquial name is so frequently used in the scientific literature that we cannot completely ignore it, especially when σ_i_ is strictly defined for external conditions. Precise calculations of magnetic shielding are usually performed for isolated molecules, and it is expected that NMR experiments can also deliver similar results for comparison. Practically, it is possible only for NMR measurements in the gas phase, and for this purpose, fluoromethanes (CH_4-n_F_n_) are really a very good group of chemical compounds that allow performing the precise verification of calculated NMR spectral parameters in isolated molecules. 

All fluoromethanes (CH_4-n_F_n_) are gaseous compounds at room temperature. They can contain three different magnetic nuclei with a one-half nuclear spin and various natural abundances: ^1^H 99.9885%, ^13^C 1.07%, and ^19^F 100% [26]. Methane (CH_4_) is included here just for the comparison of variable NMR parameters upon the following substitution of fluorine atoms. The three molecules of fluoromethanes are polar, having permanent dipole moments [32]: CH_3_F 1.858 ± 0.002 D, CH_2_F_2_ 1.9785 ± 0.01 D, and CHF_3_ 1.6515 D. It is important because these polar molecules can strongly interact even in the gas phase due to intermolecular hydrogen bonds [33,34,35]. In NMR spectra of fluoromethanes, there is the possibility of observing the distinct indirect spin-spin couplings across one and two chemical bonds: ^1^J(^1^H-^13^C), ^1^J(^13^C-^19^F), and ^2^J(^1^H-^19^F), respectively. The absolute values of spin-spin coupling constants are directly measured from NMR spectra, and they are generally less dependent on medium effects than shielding, but sometimes, this dependence is well marked, as it was shown for ^1^J(^19^F-^29^Si) in SiF_4_ [36] or ^1^J(^19^F-^33^S) in SF_6_ [37]. In the present review, all the spin-spin couplings in fluoromethane molecules are discussed in a similar way as shielding parameters.

## 2. NMR Spectral Parameters of Isolated Molecules

### 2.1. Chemical Shifts and Magnetic Shielding

NMR spectra are mostly observed for liquids, and chemical shifts are measured relative to the reference signal as defined by Equation (1). It shows that we can easily obtain an unknown shielding constant (σ_i_) if the reference value (σ_ref_) is available. In practice, this procedure is quite complex due to medium effects that can influence both the shielding parameters (σ_i_ and σ_ref_) in Equation (1) [15,38,39,40,41]. It was clear that smaller medium effects could be still present in nuclear shielding even in the gas phase [17]. Thus, there is a question of how can we obtain the most accurate shielding in isolated molecules. As shown by Raynes, Buckingham, and Bernstein [42], this problem could be solved by extrapolating ^1^H shielding of gaseous molecules to the zero density point and explaining it as follows: (2)$$\sigma^{A\;}{=\sigma }_{0}^{A}{+\sigma }_{1}^{A}\rho_{A}{+\sigma }_{2}^{A}\rho_{A}^{2}+\ldots ,$$
where σ^A^ is the nuclear magnetic shielding of A molecule in the gas phase; σ_0_^A^ represents the same shielding at zero density, which is equivalent to the value of an isolated molecule; and ρ_A_ is for gas density. σ_1_^A^ and σ_2_^A^, are the second and third coefficients of the above expansion (2), and they are parameters dependent on A-A bimolecular and A-A-A trimolecular collisions, respectively. Let us note that the linear dependence of shielding on gas density is usually observed in NMR spectra for low-density measurements. Then Equation (2) is simplified to the following:(3)$$\sigma^{A}{=\sigma }_{0}^{A}{+\sigma }_{1}^{A}\rho_{A}$$
and can be extended even on gas solutions if a small amount of compound A is observed in another B gas, which is used as a solvent [43]:(4)$$\sigma^{A}{=\sigma }_{0}^{A}{+\sigma }_{1}^{\mathrm{AB}}\rho_{\mathrm{AB}}$$

In the last equation, σ_1_^AB^ describes A-B molecular interactions, and ρ_B_ is the gas solvent density. Due to Equations (1)–(3) we are able to determine the nuclear magnetic shielding in an isolated molecule, σ_0_^A^, which is also known as the shielding constant. This parameter is still dependent on temperature and is usually determined at the fixed standard temperature, 300 K [44]. 

### 2.2. Referencing of Shielding

There is one more problem with NMR measurements in gases: the shielding of the reference compound (TMS or its solution in CDCl_3_) must be known with satisfactory precision when Equation (1) is applied for the determination of σ^A^ (gas). Fortunately, the shielding parameters in single TMS molecules (σ_ref_) are already well known: 30.783(5) ppm for ^1^H, 188.04(10) ppm for ^13^C, and 376.4(20) ppm for ^29^Si [45]. In this, the σ_ref_ data are also established for liquid TMS and for 1% TMS solution in liquid CDCl_3_. It may be interesting that the above standard shielding values are already not so crucial in NMR experiments because we can alternatively use a new universal shielding standard suitable for all the magnetic nuclei: an isolated ^3^He atom [46], for which its nuclear magnetic shielding was calculated with great precision, i.e., σ_0_(^3^He) = 59.96743 ppm [47]. Combining this result with the new values of nuclear magnetic moments [48,49] and the measurements of absolute resonance frequencies, the direct measurement of nuclear magnetic shielding in NMR spectra σ_X_ is possible according to the following relationship [50,51]:(5)$$\sigma_{X}=1\;-\;\frac{\nu_{X}}{\nu_{\mathrm{He}}}\cdot \frac{\left|\mu_{\mathrm{He}}\right|}{\left|\mu_{X}\right|}\cdot \frac{I_{X}}{I_{\mathrm{He}}}\left({1\;-\;\sigma }_{\mathrm{He}}\right),$$
where ν_X_ and ν_He_ are the absolute resonance frequencies of X and ^3^He samples, and µ_X_ and µ_He_ the nuclear magnetic moments of X and ^3^He nuclei together with their spin numbers I_X_ and I_He_, respectively. Let us add that the shielding parameters can be transferred from the ^3^He atom on liquid-deuterated solvents [51,52], so there is no need for the use of gaseous helium-3 in every NMR measurement of shielding. It makes this method of shielding measurement easy because deuterated solvents are usually applied for the lock system of standard NMR spectrometers. Knowing the absolute resonance frequencies ν_i_ measured by the NMR spectrometer, one can determine the absolute shielding of many various nuclei [50,51,52]. As seen, the reading of nuclear magnetic shielding is easily available, and the precise comparison of experimental and calculated values can be performed at least for small molecules. 

### 2.3. Indirect Spin-Spin Coupling

Between magnetic nuclei in a molecule, spin-spin coupling occurs, which is observed as the splitting of NMR signals. In contrast to shielding, spin-spin couplings do not require any reference standard, and their absolute values ^n^J_AB_ are often available directly from NMR spectra. ^n^J_AB_ means the spin-spin coupling constant via *n* chemical bonds [14], but again, the ^n^J_AB_ parameter is the averaged value of the second-rank tensor, and it is never constant for the same reason as the shielding parameter σ. Spin-spin couplings are also modified by intermolecular interactions [53] and by temperature though such effects in ^n^J_AB_ parameters are less significant than in shielding. Similarly, the measurements of ^n^J_AB_ in gases can be performed, and the results can be studied for example by exploring the following equation:(6)JnAB=(JnAB)0+(JnAB)1ρA
and one can determine the indirect spin-spin coupling constant in an isolated molecule, (^n^J_AB_)_0_ [36,54]. Fluoromethanes contain only the nuclei with the spin I = ½ (^1^H, ^13^C, and ^19^F), and their NMR spectra give fairly sharp lines due to the relatively long lifetimes of appropriate magnetic states. It allows the reading of such spin-spin coupling constants with high precision. In gases at low pressure, the lifetimes are considerably shortened, and the observation of ^n^J_AB_ is less precise due to the efficient spin-rotation mechanism of relaxation. However, the measurements of ^n^J_AB_ performed according to the idea of Equation (5) deliver usually well-marked linear dependence on density, which gives a more precise reading of (^n^J_AB_)_0_. This result can be additionally confirmed by observing spectra from two different nuclei, namely A and B. In the next section, we present the comparison of experimental and calculated values for both the NMR spectral parameters, i.e., shielding σ_0_ and spin-spin coupling ^n^J_0_ values in isolated molecules.

## 3. Experimental vs. Calculated NMR Spectral Parameters

Calculations of NMR shielding and spin-spin coupling in a molecule are rather complex, and they always consist of a few steps. The first step belongs to the optimization of molecular geometry at the lowest possible energy level (equilibrium geometry), and the calculation of a spectral parameter (σ_0_ or ^n^J_0_) is the next task. The treatment of electron correlation is certainly the most important factor at the latter level. Let us also add that four different mechanisms of spin-spin coupling must be taken into account, and therefore, the calculations of shielding are always a bit easier than the modeling of spin-spin coupling. After completing the calculation of the spectral parameter (σ_0_ or ^n^J_0_) at the equilibrium geometry, one must add the corrections for molecular vibrations: they are called ZPV (zero point vibration) corrections at the lowest vibrational level. Further corrections are required for the rovibrational motion of a molecule with the temperature increase and also for the relativistic effects in molecules containing heavy atoms, i.e., for atoms with Z < 36 [31,55,56]. Altogether, the calculations of NMR spectral parameters are difficult and time-consuming. Varieties of different calculation schemes are applied to solve the above problems, and they are summarized in numerous review publications, e.g., ref. [29,30,31]. Looking for the most accurate calculations, we should consider first of all the modern ab initio GIAO (Gauge Included Atomic Orbitals) methods proposed by Gauss and Stanton and named the coupled-cluster calculations [57,58], cf. CCSD (Coupled Cluster Singles and Doubles) or CCSD(T) (Coupled Cluster Singles and Doubles with Perturbative Triples Corrections) methods. They can deliver very accurate results of NMR spectral parameters but can be applied to relatively small, isolated molecules. At present, large molecular objects are theoretically studied rather by DFT (Density-Functional Theory) methods, which use approximate potentials. They are really efficient, but their results for small molecules are not as accurate as the others from state-of-the-art GIAO methods [59]. DFT calculations are extremely effective when contracted basis sets are specially designed for the calculations of shielding [60] or spin-spin coupling constants [61] and obviously require less time for computing. Considerably easier are also calculations of NMR spectral parameters for molecules in solids and liquids with the use of the GIPAW (Gauge Including Projector Augmented Waves) method [62]. Calculations of indirect spin-spin coupling constants are much more complex than the modeling of shielding and often require other than CCSDT methods based on MCSCF (Multi-Configurational Self-Consistent Field) or SOPPA (Second-Order Polarization-Propagator Approximation) models [30,63] and other basis sets, as mentioned above [61].

### 3.1. Chemical Shifts

Modeling of chemical shifts by quantum chemistry methods is very popular and useful in chemistry. It is due to the significance of NMR in the chemical analysis of organic compounds. According to Equation (1), the theoretical value of chemical shift can be precisely obtained when two shielding constants (σ_ref_ and σ_i_) are determined with sufficient accuracy. Let us verify this idea with a modest example taken from the literature on calculated ^19^F chemical shifts of fluoromethanes [64]. Table 1 displays the fluorine chemical shifts that were calculated using the DFT (B3LYP-GIAO) method and verified with ^19^F NMR measurements performed for pure liquids. Note that the results cited here are not complete and represent only the use of two basis sets from ref. [64]: 6-31G(d,p) and 6-31G++(d,p), but the same problem exists in the majority of other chemical shift calculations.

The two columns of chemical shifts are obtained with the above-mentioned basis sets, while the last one presents experimental chemical shifts. It is not bad if we check the calculated 31G++(d,p) chemical shifts with the experimental shifts determined for pure liquids [64], but let us verify it with Equation (1) and the σ_ref_ value in there. The numerical values of CFCl_3_ reference are given in Table S2 of the Supplementary Materials attached to the original work [64] and shown in the last row of our mini-table. The calculated chemical shifts of fluoromethanes seem better than the direct comparison of calculated shielding available for CFCl_3_. Probably, many inaccuracies of shielding calculations were accidentally canceled in the final results of fluoromethane chemical shifts. However, on the basis of such calculated ^19^F chemical shifts, we can properly assign NMR signals to particular chemical compounds. It looks as though the approximate calculations can be still useful in analytical practice, but their accuracy remains uncertain. It is quite common in the literature that some DFT-estimated chemical shifts have unusually high precision, which cannot be obtained for shielding constants, cf. ^19^F chemical shifts relative to SiF_4_, as presented in ref. [65]: calc. −107.96 ppm vs. exp. −107.71 ppm for CH_3_F, 88.40 vs. 89.06 ppm for CHF_3_, and 103.87 vs. 104.21 ppm for CF_4_. As seen, the accuracy of these calculated chemical shifts must be rather artificial because the experimental σ_ref_ value of SiF_4_ is known only with ±6 ppm accuracy from the ^19^F measurement of HF and SiF_4_ gases at the zero-density limit [66]. The numerical values of σ_ref_(SiF_4_) are not given in ref. [65].

As mentioned, the calculated chemical shifts can be useful in chemical analysis, and therefore, their applications are widely discussed in numerous review articles [55,67,68,69,70,71,72], from the semiempirical IGLO (Individual Gauge for Localized Orbitals) approximation [67] to the modern methods with relativistic corrections to shielding [55,71,72]. The most recent studies in this field have been reviewed by Kupka [73,74]. Below, we come back to the problem of accuracy in the calculated nuclear magnetic shielding and indirect spin-spin coupling parameters. Having reliable shielding constants, we are always able to determine also reliable chemical shifts according to Equation (1).

### 3.2. Nuclear Magnetic Shielding and Indirect Spin-Spin Coupling

Shielding is the most sensitive parameter of NMR spectra; its measurement is possible [50,51,52] but only with limited precision. The same is true for indirect spin-spin coupling constants. Their absolute values can be obtained from NMR spectra with likewise limited precision. Moreover, the sign of coupling constant must be determined using, for example, a double NMR resonance experiment [75] or calculated with the advanced quantum-chemical method [76,77]. Figure 1 shows the correct way of comparison between the experimental and calculated NMR parameters for molecules: shielding (σ) and spin-spin coupling (^n^J). First, let us consider only those nuclei with the one-half spin number (I = ½) in liquids: their lifetime on magnetic levels is long, and the observation of the natural line-with of their NMR signals is possible; then, the sharp signals allow for the most precise determination of the signal positions. It is represented by Level 2 in Figure 1, where the precision of measurement can be limited only by Heisenberg’s uncertainty principle [78]. At present, the exact comparison of experimental and calculated parameters (σ_liq_ and J_liq_) on Level 2 is practically impossible because it requires giant quantum mechanical calculations for all possible interactions between molecules in the liquid state. NMR signals in liquids are significantly modified by the medium effects and by temperature. Evaluation of the medium effects in liquids on shielding [21] and spin-spin coupling [53] is possible using the model of reaction field, but it is only the estimation with low accuracy. Level 1 regards the NMR measurements of gaseous compounds, and the precision of reading signals (σ_gas_ and J_gas_) is lower on Level 1 than on Level 2 because of the fast rotation of molecules in gases. However, it is a step in the right direction; the medium effects in the gas phase are considerably smaller than in liquids, and the measurement of the molecular σ parameter in a molecule is more accurate. The studies of shielding as the function of gas density permit the extrapolation of results to the zero-density point, and this way, we can gain shielding and spin-spin constants for isolated molecules (σ_0_ and J_0_) at Level 0. It was first done for the shielding of protons [42], next for ^19^F [43], and a bit later also for ^13^C [79]. A similar procedure was applied for the observed spin-spin couplings in gases, and the J_0_ parameters were determined for many gaseous compounds [36,54]. The precision of σ_0_ and J_0_ reading seems to be better than for the single measurements of σ_gas_ and J_gas_ values, but one should remember that it is true only when all the experimental procedures are perfectly completed. Moreover, the σ_0_ and J_0_ parameters are extrapolated but not directly observed. It is better to assume that the precision of their determination is considerably limited, as shown in Figure 1 at Level 0.

Levels 1 and 2 in Figure 1 are exclusively reserved for the theoretical results of σ_0_ and (^n^J_AB_)_0_ parameters. Their values can be calculated first for the equilibrium geometry: (eq) on Level 2. Adding the vibrational corrections Level 1 is reached, and the calculated parameters have the σ_0_(ZPV) and J_0_(ZPV) values equivalent to exactly their state at 0 K temperature. The precision (not the accuracy) of calculations is slightly lower on Level 1 than on Level 2 because the calculations are just more complex. The result of Level 1 still requires improvements because the NMR measurements needed for comparison cannot be performed in very low temperatures. The next correction is needed for temperature effects, which are due to the rovibrational motion of molecules in gases. Finally, Level 0 is also reached for the calculations where the experimental and theoretical data can be compared at the standard temperature, usually at 300 K. The complete procedure of such a comparison is not easy because both the experimental and calculated results introduce some error bars, as shown in Figure 1.

Below, we present the comparison of experimental and theoretical σ_0_(300 K) and J_0_(300 K) values for the choice of CH_4-n_F_n_ molecules in Table 2 and Table 3. We have tried to select only the best available results in order to verify the quality of the comparison if selected shielding and spin-spin coupling constants come from different laboratories but are limited to the same group of molecules. We hope that the choice of CH_4-n_F_n_ compounds is suitable for our task.

## 4. Experimental and Calculated Spectral Parameters for Isolated CH_4-n_F_n_ Molecules

### 4.1. ^1^H, ^13^C, and ^19^F Nuclear Magnetic Shielding

Table 2 presents selected experimental σ_0_(exp) and calculated σ_0_(calc) results for ^1^H, ^13^C, and ^19^F shielding constants observed for isolated CH_4-n_F_n_ molecules. The measured results are from NMR experiments performed in the gas phase and extrapolated to the zero-density limit. The calculated shielding constants are chosen from a great number of calculations, and only the most advanced data are placed in Table 2. As seen, it is not a review of all the available results but only the selection of reliable data according to our best knowledge. Extrapolated shielding constants (σ_0_(exp)) are consistent within experimental error bars for the studied nuclei in CH_4-n_F_n_ molecules. Note that the measurements come from many various laboratories [24,79,80,82,84,85,88,89,91,92,95,96,97], and it proves that we have reliable experimental ^1^H, ^13^C, and ^19^F shielding constants. This result is graphically presented in Figure 2, where the sequential substitution of CH_4_ by strongly electronegative (4.0) fluorine atoms leads to the decrease of all the observed nuclei, including ^19^F as well.

In contrast, the calculated shielding constants (σ_0_(calc)) are not as consistent as the experimental parameters. It is due to different methods of calculations applied in the preparation of σ_0_(calc) values. We have tried to place the best results in the first lines for each molecule, and it is easy to check that the data come mostly from the advanced quantum-mechanical calculations, usually from CCSD, CCSD(T), or CCSDT methods with large basis sets. The method of each calculation can be verified in the original publications, but it is also well marked in Table 2. For example, we can observe the obvious discrepancy between the theoretical and experimental data for proton shielding in the CH_4_ molecule, but it points out the lack of the rovibrational correction in proton shielding. Generally, this discrepancy of 1 ppm only in the proton shielding must be considered rather large; ±1 ppm in ^13^C or ^19^F shielding is fairly meaningless. Thus, for the rest of the presented data, the importance is the excellent agreement between the most advanced calculations and NMR measurements of σ_0_(300 K) shielding constants. It proves that the applied methods of shielding calculations are fantastic, and they can deliver reliable results.

### 4.2. ^1^J_CH_, ^1^J_CF_, and ^2^J_FH_ Indirect Spin-Spin Coupling Constants

Table 3 contains the experimental and theoretical spin-spin coupling constants for CH_4-n_F_n_ molecules. We can observe similar consistency of experimental data from various laboratories if the results were determined for isolated molecules and, generally, also good modeling of spin-spin coupling constants. It is incredible because the calculations of spin-spin couplings are much more complex than shielding modeling. Therefore, some calculation methods are mixed, and such a mixture delivers extraordinarily good results, cf., for example, ref. [87]. The authors optimized molecular geometry for investigated molecules with the use of the DFT method and completed the rest of the calculations applying CCSD, CCSD(T), and CCSDT advanced modeling. The latter results of shielding and spin-spin couplings (shown in Table 2 and Table 3) are really very good and only a bit worse than the complete state-of-the-art calculations [99,100]. Coming back to Table 3, we note that the comparison of calculated and measured spin-spin coupling constants is good in the case of ^1^J_CH_ but a little worse for ^1^J_CF_. One can presume that it is the influence of electronegative fluorine atom, but the consistency between experimental and calculated ^2^J_FH_ is again almost ideal in every case. It may also be an effect of the fluorine atom, which lasts only across one chemical bond. The experimental results of Table 3 are presented graphically in Figure 3. As shown, the spin-spin coupling constants of ^1^J_CH_ and ^2^J_HF_ in isolated molecules are increased with the substitution of fluorine atoms. However, negative values of ^1^J_CF_ do not further maintain this dependence by increasing for CF_4_. It is a really interesting case of spin-spin coupling that was first observed in the gas phase already 45 years ago [24]. Actually, it is well confirmed by numerous advanced calculations, as seen in Table 3.

## 5. Conclusions

It is remarkable how the new magnetic properties of atomic nuclei that were discovered from physical experiments, considered exotic at the time [1,2], quickly became a fundament for the new spectroscopy method [3,4]. Currently, this NMR spectroscopy is one of the most important techniques used in numerous fields in physics, chemistry, biology, and medicine (MRI, magnetic resonance imaging). The whole development of NMR can occur during one person’s lifetime, especially if we consider the period since the first NMR spectra were obtained [6,7].

Due to the modern development of computing systems, we can model the σ_0_ and J_0_ spectral parameters. As shown in this review, the present results for CH_4-n_F_n_ molecules from various laboratories are quite consistent and can be used for the verification of calculated shielding and spin-spin coupling constants. The quality of the theoretical NMR parameters depends on the applied method of calculations, and in the near future, we can certainly expect further progress in this field. This progress should lead to the exact modeling of molecular interactions in condensed phases, i.e., liquids and solids. For this purpose, one must investigate molecular interactions between two molecules spatially orientated to average the results gained from their different orientations. The first steps of such studies were completed already in 1978 [105,106]. Nowadays, the accurate treatment of this subject is possible but still time-consuming, as shown for the shielding in the N_2_ pair of molecules [107].

The results presented in Table 2 and Table 3 confirm the capability of the precise modeling of shielding and spin-spin coupling constants by ab initio methods. The calculated results are good in every case but are not perfect. Some effects in these calculations were fully neglected, e.g., relativistic corrections. They are probably quite small even for shielding in fluorine atoms (in a range of 1 ppm), but they exist. On the other hand, the calculated results of ^19^F shielding are more accurate than the experimental data, which give the absolute shielding with the error bar ±6 ppm [66]. This is due to the difficulty of experimental work with extremely aggressive hydrogen fluoride (HF) gas. Most accurate NMR measurements are carried out in glass, but it is not possible for hydrogen fluoride. It is worth underlining that such good calculations of ^19^F shielding in CH_4-n_F_n_ molecules shown in Table 2 permit us to limit the error bar in the absolute scale of ^19^F shielding to the range of ±2 ppm, which is consistent with earlier estimation obtained by DFT calculations [65]. Altogether, the choice of CH_4-n_F_n_ molecules for the verification of the present capability of theoretical prediction of multinuclear shielding and spin-spin coupling seems to be good because it gives us insight into the accuracy of state-of-the-art calculations of spectral NMR parameters for isolated molecules.

## Figures and Tables

**Figure 1 molecules-28-01499-f001:**
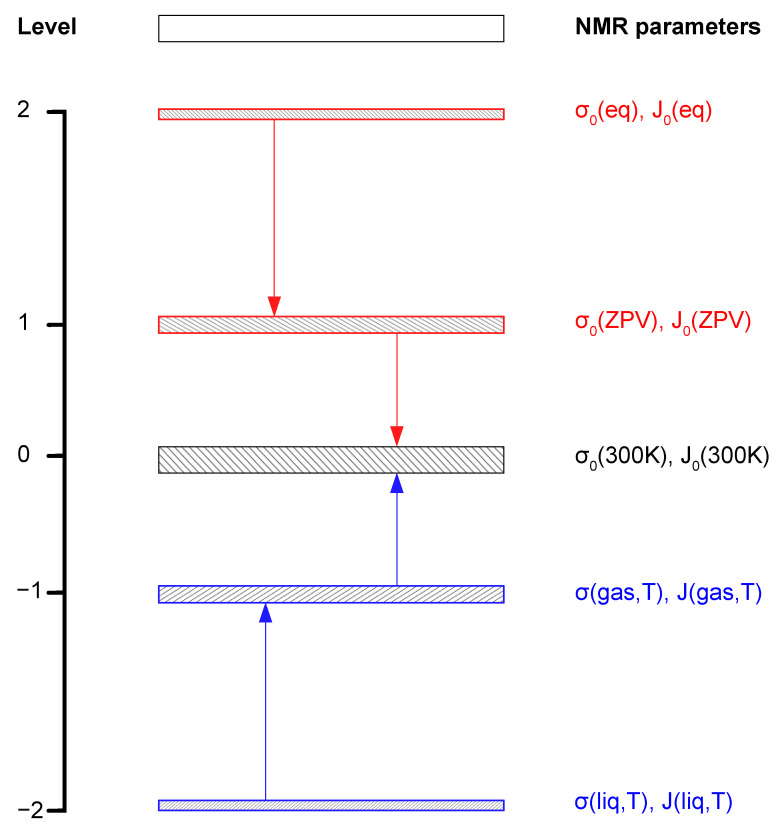
Comparison of experimental and calculated NMR spectral parameters σ and nJ. The thickness of Levels illustrates the increasing uncertainty of determining parameters from −2 to 0 for experiments and from 2 to 0 for calculations.

**Figure 2 molecules-28-01499-f002:**
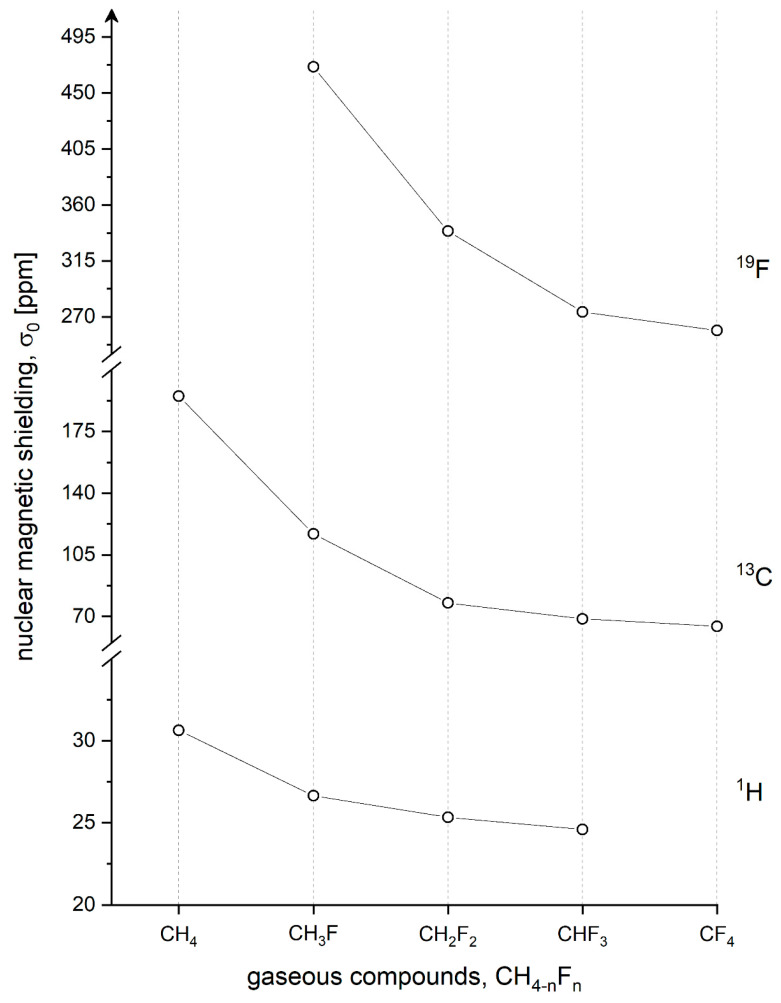
Experimental ^19^F, ^13^C, and ^1^H NMR shielding values in isolated molecules of fluoromethanes. Each substitution of an electronegative fluorine atom leads to significant deshielding effects for all the nuclei in fluoromethane molecules. Note: The scale of shielding is different for ^19^F, ^13^C, and ^1^H nuclei for better visualization.

**Figure 3 molecules-28-01499-f003:**
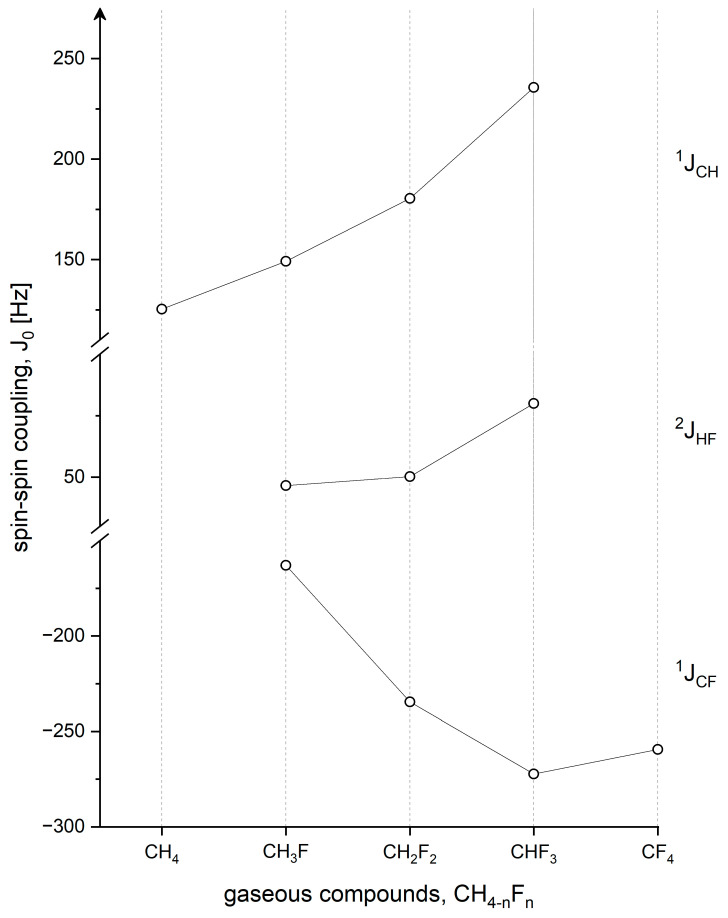
Spin-spin couplings of fluoromethanes without the influence of intermolecular interactions as seen in ^19^F, ^13^C, and ^1^H NMR measurements. The opposite signs of ^1^J_CH_ and ^1^J_CF_ coupling constants were first found in double-resonance experiments [104].

**Table 1 molecules-28-01499-t001:** An example of the GIAO-DFT calculated ^19^F chemical shifts for fluoromethanes; the selected data are cited from Ref. [64].

Molecule	6-31G(d,p)	6-31G++(d,p)	Experiment
CH_3_F	−278.22 ppm	−279.16 ppm	276.3 ppm
CHF_3_	−93.91 ppm	−87.13 ppm	−84 ppm
CF_4_	−80.71 ppm	−70.43 ppm	−69 ppm
*CFCl_3_*	*179.41 ppm*	*179.16 ppm*	*188.7 ppm*

**Table 2 molecules-28-01499-t002:** Selected shielding constants for CH4-nFn isolated molecules available from gas-phase NMR experiments ^a^ and advanced ab initio calculations of quantum chemistry ^b^.

Parameter	^1^H Shielding (ppm)		^13^C Shielding (ppm)		^19^F Shielding (ppm)	
Molecule	σ_0_ ^ H ^ (exp)	σ_0_^H^ (calc)	σ_0_ ^ C ^ (exp)	σ_0_^C^ (calc)	σ_0_ ^ F ^ (exp)	σ_0_^F^ (calc)
CH_4_	30.633(6) ^c^ 30.611(24) ^g^	31.41 ^d^ 31.6 ^q^ 31.4 ^w^ 31.93 ^z^	195.01 ^e^ 195.1 ^h^ 195.15 ^i^	195.2 ^j^ 196.2 ^f^ 193.6 ^d^ 197.52 ^z^		
CH_3_F	26.635(8) ^c^ 26.62 ^l^	26.59 ^k^ 26.90 ^l^ 27.52 ^d^	116.83 ^l^ 116.69 ^o^ 116.8 ^h^ 116.3 ^s^	118.30 ^k^ 118.8 ^f^ 122.0 ^d^ 120.3 ^l^	470.98(1) ^m^ 471.0 ^p^	472.9 ^n^ 475.85 ^r^ 482.0 ^d^
CH_2_F_2_	25.331(3) ^t^	25.24 ^k^ 26.58 ^d^	77.726(4) ^t^ 77.6 ^s^	78.87 ^k^ 89.6 ^d^	338.935(2) ^t^ 339.1 ^u^	340.7 ^n^ 358.40 ^r^ 363.7 ^d^
CHF_3_	24.583(10) ^c^	24.55 ^k^ 25.80 ^d^	68.738(8) ^v^ 68.5 ^s^	69.28 ^k^ 82.6 ^d^	273.988(4) ^v^ 274.1 ^p^	275.2 ^n^ 298.03 ^r^ 302.0 ^d^
CF_4_			64.5 ^h^ 64.5 ^s^	63.9 ^l^ 79.1 ^d^	258.80 ^o^ 259.0 ^u^	259.5 ^n^ 278.0 ^d^ 281.22 ^r^

^a^ Extrapolated to the zero-density point and otherwise as marked; ^b^ GIAO calculations as described; ^c^ ref. [80]; ^d^ HF (Hartee–Fock), ref. [81]; ^e^ ref. [82]; ^f^ MP2 (Møller–Plesset Perturbation Theory (second order)) including ZPV, ref. [83]; ^g^ ref. [84]; ^h^ ref. [85]; ^i^ ref. [79]; ^j^ CCSD(T) ref. [58], including −3.695 ppm for the rovibrational correction given in ref. [86]; ^k^ DFT and CCSD(T) jointly, ref. [87]; ^l^ ref. [88]; ^m^ ref. [89]; ^n^ CCSD(T) including ZPV and temperature corrections, ref. [90]; ^o^ ref. [91]; ^p^ ref. [92]; ^q^ CCSD(T) ref. [58]; ^r^ HF, including ZPV, Ref. [93]; ^s^ low-density gas without extrapolation, ref. [24]; ^t^ ref. [94]; ^u^ ref. [95]; ^v^ ref. [96]; ^w^ MP2, ref. [97]; ^z^ HF, ref. [98].

**Table 3 molecules-28-01499-t003:** Selected indirect spin-spin coupling constants for CH_4-n_F_n_ isolated molecules available from gas-phase NMR experiments ^a^ and advanced ab initio calculations of quantum chemistry ^b^.

Parameter	^1^J(^13^C−^1^H) Coupling (Hz)		^1^J(^13^C−^19^F) Coupling (Hz)		^2^J(^19^F−^1^H) Coupling (Hz)	
Molecule	^ 1 ^ J_0_(exp)	^1^J_0_(calc)	^ 1 ^ J_0_(exp)	^1^J_0_(calc)	^ 2 ^ J_0_(exp)	^2^J_0_(calc)
CH_4_	125.304 ^c^ 125.31 ^d^	125.65 ^e^ 124.26 ^f^ 120.61 ^g^				
CH_3_F	147.37(5) ^h^ 149.15 ^i^	148.09 ^g^ 145.62 ^f^ 141.5 ^k^	−163.10(5) ^h^ −163.00(2) ^k^ −160.2 ^l^	−160.8 ^f^ −156.6 ^k^	46.64(5) ^h^	46.81 ^f^ 46.3 ^k^
CH_2_F_2_	180.42(5) ^m^ 180.38(4) ^k^	179.85 ^f^ 175.7 ^k^	−234.55(5) ^m^ −233.91(11) ^k^ −232.7^l^	−224.52 ^f^ −220.7 ^k^	50.24(5) ^m^	50.25 ^f^ 51.9 ^k^
CHF_3_	235.63(5) ^n^ 235.26(9) ^k^	235.63 ^f^ 236.8 ^k^	−272.29(5) ^n^ −272.18(7) ^k^ −272.4 ^l^	−238.28 ^f^ −242.1 ^k^	79.92(5) ^n^	79.18 ^f^ 79.3 ^k^
CF_4_			−258.32(9) ^h^ −259.4 ^l^	−272.84 ^g^		

^a^ Extrapolated to the zero-density point and otherwise as marked; ^b^ ab initio calculations or as described; ^c^ ref. [99]; ^d^ ref. [100]; ^e^ CCSD, ref. [100]; ^f^ DFT and CCSD(T) jointly, ref. [87]; ^g^ MP2 or experimental geometry and CCSD, ref. [101]; ^h^ ref. [91]; ^i^ ref. [102]; ^k^ MCSCF LR (Multi-Configuration Self-Consistent-Field Linear Response), ref. [103]; ^l^ low-density gas without extrapolation, ref. [24]; ^m^ ref. [94]; ^n^ ref. [96].

## Data Availability

No new data were created or analyzed in this study.

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
