# Peer review of "Searching for the Best Values of NMR Shielding and Spin-Spin Coupling Parameters: CH4-nFn Series of Molecules as the Example"

_molecules, 2023, doi:10.3390/molecules28031499_

Round 1
Reviewer 1 Report
This manuscript reviews recent advances in computation and experimental measurements of NMR shielding constants (chemical shifts) and spin-spin coupling constants involving fluorine in the series of fluoromethanes. It is very well known that fluorine NMR parameters are very capricious in the sense of their experimental measurement and this is much more so from the computational point of view. Authors of the present review are the well-known specialists in this field, especially in the experimental measurements including those in gas phase and even in dilute gas extrapolated to the zero-density limit. This review, when published, will be of major practical interest to physical chemists and NMR spectroscopists working in this field, so that I have no doubt to recommend it for publication in its present form.
Author Response
Dear Reviewer,
Thank you for your positive opinion of our manuscript entitled: “Searching for the Best Values of NMR Shielding and Spin-spin Coupling Parameters. CH4-nFn Series of Molecules as the Example”, we appreciate it very much.
Sincerely yours,
Karol Jackowski and Mateusz Słowiński
Reviewer 2 Report
The review article concerns the comparison between the theoretical and experimental parameters of the NMR spectrum, particularly on the shielding and spin-spin coupling constants of the CH4-nFn molecules. The manuscript presents some interesting aspects of these properties, mostly on the difficulties in obtaining accurate values for isolated molecules at room temperature. The manuscript is well-written and can be accepted for publication after the authors consider the following points.
- I understand that the GIAO method should be the focus of the text. However, the GIPAW also must be cited, at least to refer the reader to this possibility. This method is an alternative for periodic condition calculations of the shielding constants.
- I agree that the choices of the DFT functionals or pos-HF methods are fundamental for accurate results for shielding and spin-spin coupling constants. However, the manuscript only superficially comments on the importance of the basis sets. Thus, the bases specifically developed for shielding and spin-spin coupling constants calculations must be cited and discussed throughout the text. I highlight the basis sets family of F. Jensen [J. Chem. Theory Comput. 2015, 11, 132−138, Theor Chem Acc (2010) 126:371–382].
- The relativistic effects appear only in the conclusion section. These effects can be considered by using ZORA or DKS methods, for example, and these possibilities must be cited and discussed throughout the manuscript.
- The article “Progress in Nuclear Magnetic Resonance Spectroscopy, 108, (2018) 17-73” and the book “ Calculation of NMR and EPR Parameters: Theory and Applications" by Martin Kaupp, Michael Bhl, and Vladimir G. Malkin should be cited.
- On page 6, line 247, one has; “... or calculated with the advanced quantum-chemical method.” Please, cite the references here.
- In Table 2, some values were calculated at a low level of theory, HF for example. The coupled-cluster values obtained in reference 56 of the manuscript must be included in Table 2
- On page 6, lines 363 to 367. The text needs to be clarified, it is a bit confusing.
- The first two paragraphs of the Conclusion section deserve more attention from the authors.
Author Response
Dear Reviewer,
Thank you for your review report on our manuscript entitled: “Searching for the Best Values of NMR Shielding and Spin-spin Coupling Parameters. CH4-nFn Series of Molecules as the Example”. We have tried to fill out your requirements as follows:
Ad.1. Information on the GIPAW method is included in the new version of our manuscript, lines: 204-205 and Ref. [104]
Ad.2. Jensen’s basis sets are mentioned in our manuscript (lines 200-203) and cited as Ref. [102,103].
Ad.3. We could not find any relativistic effects estimated for shielding and spin-spin coupling constants in CH4-nFn molecules. It seems that the relativistic effects are of secondary importance in this case. They are obviously significant for heavier nuclei but not so much for our comparison. Anyway, we added comments on this subject in lines 187-189 and Ref. [31,67,101].
Ad.4,5. Ref. [106,107] are included according to your suggestion.
Ad.6. We are really very thankful for your comment on Ref.[56]. The new value of 13C shielding in methane (195.2 ppm) is the best among the other results. We have only included the rovibrational correction for 300K given by Raynes et al., Ref. [108].
Ad.7,8. The pointed-out parts of our manuscript are improved in the new text.
Regards,
Karol Jackowski and Mateusz Słowiński
Reviewer 3 Report
In this mini-review, the authors carried out a very thorough and interesting chronological analysis of the literature sources dealing with the calculated and experimental data of the NMR spectral parameters of methane and its fluorine-substituted derivatives. I recommend publishing this review as it is.
P.S. It would like to see similar review(s) from the authors on other halogen-derivatives of methane (Cl, Br, I).
Author Response
Dear Reviewer,
Thank you very much for your positive review of our manuscript entitled: “Searching for the Best Values of NMR Shielding and Spin-spin Coupling Parameters. CH4-nFn Series of Molecules as the Example”. Your suggestion on the verification of NMR spectral parameters of heavier halogen derivatives of methane is really interesting because it would give us some insight into the relativistic corrections to shielding and spin-spin coupling constants. We will consider it in near future.
Best regards,
Karol Jackowski and Mateusz Słowiński
Round 2
Reviewer 2 Report
I recommend the article for publication in its current form.